# Adsorption of Hydrolysed Polyacrylamide onto Calcium Carbonate

**DOI:** 10.3390/polym14030405

**Published:** 2022-01-20

**Authors:** Jin Hau Lew, Omar K. Matar, Erich A. Müller, Myo Thant Maung Maung, Paul F. Luckham

**Affiliations:** 1Department of Chemical Engineering, Imperial College London, London SW7 2AZ, UK; s.lew20@imperial.ac.uk (J.H.L.); o.matar@imperial.ac.uk (O.K.M.); e.muller@imperial.ac.uk (E.A.M.); 2PETRONAS Research Sdn. Bhd., Bandar Baru Bangi 43000, Selangor, Malaysia; maungmyothant@petronas.com

**Keywords:** polymer adsorption, kinetics, adsorption isotherm, polyacrylamide, calcium carbonate

## Abstract

Carbonate rock strengthening using chemical techniques is a strategy to prevent excessive fines migration during oil and gas production. We provide herein a study of the adsorption of three types of hydrolysed polyacrylamide (HPAM) of different molecular weight (F3330S, 11–13 MDa; F3530 S, 15–17 MDa; F3630S, 18–20 MDa) onto calcium carbonate (CaCO_3_) particles via spectrophotometry using a Shimadzu UV-2600 spectrometer. The results are compared to different adsorption isotherms and kinetic models. The Langmuir isotherm shows the highest correlation coefficient (R^2^ > 0.97) with equilibrium parameters (*R_L_*) ranging between 0 and 1 for all three HPAMs, suggesting a favorable monolayer adsorption of HPAM onto CaCO_3_. The adsorption follows pseudo-second order kinetics, indicating that the interaction of HPAM with CaCO_3_ is largely dependent on the adsorbate concentration. An adsorption plot reveals that the amount of HPAM adsorbed onto CaCO_3_ at equilibrium increases with higher polymer molecular weight; the equilibrium adsorbed values for F3330S, F3530S and F3630S are approximately 0.24 mg/m^2^, 0.31 mg/m^2^, and 0.43 mg/m^2^, respectively. Zeta potential analysis shows that CaCO_3_ has a zeta potential of +12.32 mV, which transitions into negative values upon introducing HPAM. The point of zero charge (PZC) is observed at HPAM dosage between 10 to 30 ppm, in which the pH here lies between 9–10.

## 1. Introduction

Carbonate reservoirs are ubiquitous in the oil and gas industry as they contribute to approximately 60% of the global petroleum reserves and provide lucrative potential for additional gas reserves [1,2]. However, much of the global hydrocarbon reserves are often found in poorly consolidated reservoirs [3] which have a relatively young geological age. For this reason, the rock grains have not undergone sufficient natural cementation by mineral deposition; hence, they often have weak and unconsolidated structures [4]. Reservoirs with weak formation strength have many detrimental consequences to drawdown operation, especially in the form of fine solid production. As hydrocarbon is consistently extracted from the reservoir, there is a depletion in the pore pressure which causes an increase in the effective stress exerting onto the formation rock. Normally, if the rock structure is rigid and strong enough, the formation rock will only deform gradually. However, for highly porous rock or weak formation rocks, there is a high tendency of abnormal increase in effective stress with fluid withdrawal, leading to irreversible deformation [5,6,7]. As the weak rocks deform and collapse, they are crushed, leading to the production of fines, a process also known as fines migration, which may be aggravated by other factors such as high drawdown pressure, sudden collapse of formation pores and water breakthrough within reservoir [4].

In order to overcome the long-standing issue of fines production, the oil and gas industry has been developing various mitigation strategies for this purpose. These schemes can be generally categorized into either mechanical or chemical methods. Common mechanical fines entrapment techniques include stand-alone sand screens, gravel packs or resin-coated gravel packs. These techniques can also be coupled with one another to enhance the fine entrapment performance [8]. However, mechanical techniques are not without their limitations and disadvantages. These techniques are often more time-consuming and expensive when compared with chemical techniques, not to mention other issues including productivity reduction, complexity in installing the equipment, installation damage to the wellbore and causing interference to reservoir operation [9,10,11]. Therefore, chemical technique provides an attractive alternative to consolidate weak rock formation. Chemical consolidation techniques basically involve the injection of a form of reactive chemical into the loose formation to bind the loose sand grains together. The outcome of this technique is usually the increment of uniaxial compressive strength (UCS) of the formation.

Polyacrylamide (PAM), which is a commonly used polymer in Enhanced Oil Recovery (EOR), is an attractive candidate for deployment as a formation-strengthening chemical. The adsorption of PAM in oil and gas applications has been studied specifically for EOR application. Wang et al. [1] conducted an adsorption of a hydrophobic PAM onto negatively charged calcite. Their hydrophobic PAM was a copolymer of acrylamide (AM), 2-acrylamide-2-methylpropanesulfonic acid (AMPS), 2-methacryloyloxyethyl 12-alkyl dimethyl ammonium bromide (MADA) in a 1:0.35:0.05 molar ratio, and the polymer was synthesized through free radical polymerization using ammonium persulfate as initiator. According to their results, hydrophobic PAM adsorbs onto calcite via hydrogen bonding, and they exhibit a Langmuir type adsorption isotherm which indicates monomolecular PAM adsorption onto a homogeneous adsorbent. The adsorbed amount of PAM onto calcite (in terms of per gram of calcite) increases with decreasing particle size as smaller particle size corresponds to higher particle surface area. The presence of dilute salt ions also increases the amount of PAM adsorbed, as salt ions shielded the charge on the HPAM chains, contracting the polymer chains, thus enabling more polymer to adsorb. However, an increase in environment temperature decreases the adsorbed PAM amount as the formation of new surface interactions during adsorption is exothermic, thus increasing background temperature shifts the equilibrium adsorption in the opposite direction.

Peng et al. [12] studied the effect of cationic polyacrylamide (CPAM) on precipitated calcium carbonate (PCC) flocculation. They investigated the kinetics of the CPAM adsorption onto PCC, as well as the effect of CPAM charge density and background ionic strength on the adsorption. By measuring the Photometric Dispersion Analyzer Ratio (R), they were able to observe the influence of CPAM charge density and ionic strength on the final PCC flocculate size. The CPAM charge densities used were 5%, 10% and 40%, while the ionic strength was set to be 0, 0.01 and 0.1 M. They found that the adsorption behavior of CPAM on PCC is not straightforward: in the absence of salt, electrostatic attraction towards PCC dominates in high charge CPAM, while hydrogen bonding of CPAM amide group and carboxyl group of PCC dominates in low charge density CPAM. The presence of high ionic strength essentially neutralizes the charges of the polymer, thus the flocculation behavior between polymer adsorbed PCC transitions from an electrostatic attraction to physical bridging interaction. A similar study was conducted by Rasteiro et al. [13], whereby they also studied the flocculation and adsorption of CPAM on PCC. Their results agreed with that from Peng et al. [12], where in the absence of salt, an increase in CPAM charge density, adsorption equilibrium is reached at a lower contact time, indicating high charge density favors adsorption. In addition, they reported an increase in CPAM adsorption onto PCC when the molar mass of CPAM increases. They explained that lower molar mass polymer adopts a flat configuration on adsorbent surface, thus occupying more surface space. CPAM adsorption also increases when the molecular structure of CPAM is non-linear, as the introduction of side groups in the polyelectrolyte molecule transitions the flocculation mechanism from a patching to a bridging one [14,15]. Essentially, highly branched polymers produce more open flocs in the system, giving more opportunities for bridging mechanisms which lead to higher adsorption rates. In terms of study of PAM adsorption kinetics and isotherm, Zhu et al. [16] conducted a study on the adsorption of hydrolysed PAM (HPAM) onto quartz sand surface. Their results indicate that HPAM adsorption fits a Langmuir isotherm, which agrees with several published papers [1,17,18,19]. HPAM adsorption kinetics also follows a pseudo-second order model, indicating that chemisorption is the main adsorption mechanism between HPAM and quartz sand surface.

Nevertheless, to the best of our knowledge, the adsorption studies of PAM are limited to cationic PAM and sandstone, with very little reported on anionic PAM or hydrolysed PAM (HPAM) adsorption onto calcium carbonate (CaCO_3_). This is especially significant as calcium carbonate at pH below 7 is positively charged due to the presence of protonated hydrogen ions bonded to non-bridging oxygen atoms in the CaCO_3_ lattice structure [20]. It is instinctive to use an anionic polymer to perform adsorption onto a positively charged adsorbent. Therefore, this paper aims to investigate the adsorption kinetics and characteristics of HPAM onto positively charged CaCO_3_ while assessing how polymer concentration and molecular weight affect the adsorption process.

## 2. Materials and Methods

### 2.1. Material and Equipment

The HPAM used in this work were obtained from SNF Floerger (Wakefield, UK). These polymers were coded F3330S (30% hydrolysed, 11–13 MDa), F3530S (30% hydrolysed, 15–17 MDa), and F3630S (30% hydrolysed, 18–20 MDa). The molecular formula of the HPAM is illustrated in [Fig polymers-14-00405-g001], where the negative charge of the polymer comes from the deprotonation of carboxyl groups of the acrylate monomer into carboxylate group [21]. The calcium carbonate (CaCO_3_, ≥99 %) powder was purchased from VWR Chemicals Ltd. (Lutterworth, UK). The solvent used in this work was deionized water (DI, 18 MΩ Ohm). All the experiments were conducted under ambient conditions.

The experimental equipment included a 2 mag MIX15 multi-stirrer (2 mag, Munich, Germany) for sample stirring, Shimadzu UV-2600 spectrophotometer (Shimadzu, Kyoto, Japan) to study the absorbance of the sample, Anton Paar Litesizer 500 (Anton Paar GmbH, Graz, Austria) to study the zeta potential of the HPAM-CaCO_3_ mixture, a Malvern Mastersizer 2000 (Malvern, England, UK) to measure the particle size of the CaCO_3_ used, a Micromeritics TriStar 3000 (Micromeritics, Norcross, GA, USA) to determine the surface area of the CaCO_3_ used and a Fisherbrand accuSpin 400 centrifuge (Fisher Scientific, Waltham, MA, USA) to separate the two-phases of the HPAM-CaCO_3_ mixture.

### 2.2. Experimental Procedure

#### 2.2.1. Particle Size and Surface Area Determination

Before commencing the adsorption analysis, the particle size of the CaCO_3_ was first determined using the Malvern Mastersizer 2000. A total of 1 g of CaCO_3_ was dispersed into 25 mL of DI water, and the suspension was added drop by drop with a dropper into cylindrical dispersion unit of the equipment. The dispersion was added until the scattering reached the suitable obscuration range of the equipment. The average particle diameter of the CaCO_3_ used is 3.48 µm.

The surface area of the CaCO_3_ was determined by BET surface area analysis using the Micromeritics TriStar 3000. A total of 200 mg of CaCO_3_ were degassed under nitrogen at 80 °C for minimum 2 h before measuring the surface area. From the instrument, the BET surface area of the CaCO_3_ used was found to be 9.01 m^2^/g.

#### 2.2.2. Polymer Adsorption Analysis

To understand the effect of HPAM concentration on its adsorption onto CaCO_3,_ UV-Vis spectrometry of HPAM—CaCO_3_ mixtures was conducted. First, 1 g of CaCO_3_ was added into a 180 mL Bakelite cap glass bottle. After this, 25 mL of diluted polymer solutions with concentration ranging from 10 ppm to 400 ppm were prepared and poured into the Bakelite cap bottle with stirring. Each sample was triplicated to determine the standard deviation and standard error. The mixtures were subjected to 1 h, 6 h, 18 h, 24 h and 72 h stirring to determine the minimum amount of time required for equilibrium adsorption of HPAM onto CaCO_3_. After the adsorption, the mixtures were transferred into a conical bottom centrifuge tube and centrifuged at 8500 RPM for 40 min to remove the calcium carbonate particles. The supernatant was analysed using the Shimadzu UV2600 spectrophotometer at 225 nm (F3330S) and 210 nm (F3530S, F3630S) to determine the amount of HPAM adsorbed. These are the wavelengths which gives the highest relation coefficient (R^2^ > 0.99) on their respective calibration curve. The amount of HPAM adsorbed per unit surface area of carbonate, *Q_e_* (mg/m^2^) was then calculated as follows:(1)$$Q_{e}=\left(C_{0}-C_{e}\right)V/A$$
where *C*_0_ and *C_e_* are the initial and equilibrium concentration of the supernatant (mg/L), *V* is the volume of polymer used (L) and *A* is the surface area of the carbonate (m^2^). All the adsorption experiments were conducted at ambient temperature and pressure, which is 25 °C and 1 atm, respectively.

#### 2.2.3. Polymer Adsorption Kinetics

To understand the HPAM adsorption kinetics, 1 g of CaCO_3_ was added to 25 g of 300 ppm F3530S, and the mixture was stirred from 1 min to 6 h. The mixtures were collected at different time intervals and centrifuged, and the supernatant was analysed at 210 nm using the Shimadzu UV2600 spectrophotometer. The datapoints were then fitted into four different adsorption kinetic models. The first model is the pseudo-first order model [16]:(2)$$\ln\left(Q_{e}-Q_{t}\right)=\;\ln Q_{e}+k_{1}t\;\;$$
where *Q_e_* is the equilibrium PAM adsorption capacity per unit surface area of carbonate (mg/m^2^), *Q_t_* is the PAM adsorption capacity per unit surface area of carbonate (mg/m^2^) at given time *t*, and *k*_1_ is the pseudo-first order rate constant. This model assumes that the adsorption rate is directly proportional to the adsorbate concentration, is limited by the mass transfer resistance in the particle. Next, we use the pseudo-second order model
(3)$$\frac{t}{Q_{t}}=\frac{1}{k_{2}Q_{e}^{2}}+\frac{t}{Q_{e}}\;\;$$
where *k*_2_ is the pseudo-second order rate constant. This model assumes the adsorption rate is directly proportional to the square of adsorbate concentration and is limited by the adsorption mechanism. The Elovich model is next, expressed by
(4)$$Q_{t}=a+b\;\ln\;t\;$$
where *a* is the initial adsorption rate (mg/(g·min)), and *b* is desorption constant. This model assumes the adsorption energy is not uniform and increases linearly with increasing surface coverage. At the same time, adsorption rate is not uniform, but decreases exponentially with the increasing adsorption capacity. Finally, the intraparticle diffusion model is given by
(5)$$Q_{t}=k_{ip}t^{0.5}+C_{e}\;\;$$
where *k_ip_* is the rate constant of intra-particle diffusion kinetics (g/(mg·min^0.5^)) *C_e_* is the equilibrium adsorption capacity at time *t*. If the fitted curve passes through the origin, then the intraparticle diffusion is the rate controlling step of the adsorption process [22].

#### 2.2.4. Isothermal Adsorption Model Analysis

Upon obtaining the adsorption result, the experimental data points were fitted into different well-known isothermal adsorption models. The two most commonly used isotherms for adsorption experimental data fitting are the Langmuir and Freundlich isotherms [23]. The Langmuir isotherm mainly assumes a monolayer adsorption of adsorbates onto the surface of adsorbent, and maximum adsorption is achieved when the surface is completely covered. On the other hand, the Freundlich isotherm mainly assumes multilayer adsorption of adsorbates onto heterogeneous surfaces with different adsorption affinities [24]. A linear form of the Langmuir isotherm is expressed as follows:(6)$$\frac{C_{e}}{Q_{e}}=\frac{1}{Q_{m}K_{L}}+\frac{C_{e}}{Q_{m}}\;\;$$
where *Q_m_* is the maximum polyacrylamide adsorption capacity per unit surface area of carbonate (mg/m^2^) and *K_L_* is the Langmuir constant (L/mg).

An important feature of the Langmuir isotherm is a dimensionless constant which is called separation factor, or also known as equilibrium parameter (*R_L_*). It is calculated using through the following equation:(7)$$R_{L}=\frac{1}{1+K_{L}C_{0}}$$

This dimensionless separation factor is useful to evaluate how favourable an adsorption process can be. An irreversible adsorption reaction is found when *R_L_* = 0, the adsorption is favourable if 0 < *R_L_* < 1, it its linear if *R_L_* = 1, and unfavourable when *R_L_* > 1 [25].

The linear form of the Freundlich isotherm is expressed by the following equation:(8)$$\log Q_{e}=\log K_{F}+\frac{1}{n}\log C_{e}\;\;$$
where *K_F_* and 1/*n* are the so-called Freundlich constants; *K_F_* represents the relative adsorption capacity of carbonate while *n* is the degree of dependence of adsorption on the equilibrium polyacrylamide concentration.

Another interesting isothermal adsorption model to discuss is the Temkin isotherm, which is used to describe adsorption processes driven by strong intermolecular interactions, such as electrostatic interactions or ion exchange between adsorbate and adsorbent with heterogeneous surfaces [16]. We considered looking into this adsorption model as we expect the presence of an electrostatic attraction between the HPAM and CaCO_3_ which are oppositely charged. The linear form of the Temkin isotherm is expressed by
(9)$$Q_{E}=\frac{RT}{b}\;\ln K_{T}+\frac{RT}{b}\;\ln C_{e}\;\;$$
where *b* is a Temkin constant related to the heat of sorption (J·mol^−1^) while *K_T_* is the Temkin isotherm constant (L·g^−1^) [26].

#### 2.2.5. Zeta Potential Analysis

The sediments of the HPAM–CaCO_3_ mixtures after centrifuged were collected and analysed for their zeta potential. A total of 0.1 g of the residue is dispersed and diluted in approximately 10 mL of DI water before conducting the zeta potential analysis in the Anton Paar Litesizer 500. This is to ensure the conductance lies within a suitable range of values. Having too low of a conductance means the ions at the slipping layer cannot move through the solution when an electric field is applied, while having too high a conductance will cause electrolysis of the sample. The diluted sample was transferred into an Omega Cuvette which is slotted in the instrument for analysis.

## 3. Results and Discussion

### 3.1. Polymer Adsorption Analysis

The effect of polymer molecular weight was studied by plotting a curve of adsorbed amount of polymer against polymer concentration for each HPAM used. The calibration curves for each polymer are shown in Appendix A. The adsorption result of different HPAM onto CaCO_3_ surface is shown in [Fig polymers-14-00405-g002] and a table of the values is in Appendix A.

The error bars calculated from the triplicated results are incorporated into [Fig polymers-14-00405-g002]. However, the standard error is negligible, and in most cases smaller than the individual datum point. A set of triplicated results from F3330S are included in Appendix A as an example. From the curve of best plot in [Fig polymers-14-00405-g002], the adsorbed amount of both F3330S and F3530S onto CaCO_3_ increases steadily with increasing polymer concentration until approximately 150 ppm, where the adsorbed amount starts to plateau. F3630S has a more continuous increase in adsorbed amount where it only starts to plateau at approximately 300 ppm. Since the charge density of the HPAM used in all cases is 30%, they are considered rather low charge density polymers. Hence, the polymers will mainly form extended conformation with loops and tails rather than lying flat on the CaCO_3_ surface. The loops and tails are responsible for the bridging interaction between different CaCO_3_ [27]. When a high concentration of HPAM is used, the polymers are likely to compress laterally and extend further away from the particle surface. Hence, the increasing number of loops and tails only serves to hinder late-coming polymer molecules to adsorb onto CaCO_3_ surface, thus the adsorbed amount of polymer will stay constant after an optimum polymer concentration is reached [28].

As observed in [Fig polymers-14-00405-g002], F3630S, which is the HPAM with the highest molecular weight, achieved the highest equilibrium adsorbed amount onto CaCO_3_, while the equilibrium value decreases with decreasing polymer molecular weight, from F3530S to F3330S. This is in line with the findings from Rasteiro et al. [13], where they also noticed a similar pattern in cationic PAM adsorption onto CaCO_3_. Based on [Fig polymers-14-00405-g002], the slightly higher equilibrium adsorbed amount in high molecular weight polymer is attributed to the different type of polymer conformation on adsorbent surface upon adsorption. Lower molar mass polymer tends to adopt a flat configuration, with each section of molecules occupying a larger fraction of adsorbent surface. In the case of high molecular weight polymer, there is a higher likelihood of loops and tails formation on the particle surface, hence the polymers are not lying flat on the surface. The high coverage of low molecular weight polymer is more effective in hindering late-coming polymers than that in high molecular weight polymer [29]. A depiction of this phenomenon is illustrated in [Fig polymers-14-00405-g003]. In our experiment, F3330S, being lower molecular mass than F3630S, simply formed loops and tails to a lesser degree than F3630S, so it has a relatively flatter configuration than its F3630S counterpart.

### 3.2. Polymer Adsorption Kinetics

The HPAM–CaCO_3_ mixtures were stirred for 1 h, 6 h, 18 h, 24 h and 72 h to determine the suitable period of time required to achieve maximum adsorption. This investigation is imperative when it is translated to oil and gas operation, as it signifies the amount of time required for the polymer to consolidate the reservoir properly before any hydrocarbon extraction is carried out. The adsorption results of F3530S onto CaCO_3_ surface over different stirring time are summarized in Appendix A. The adsorption plot of F3530S onto CaCO_3_ over different stirring times is shown in [Fig polymers-14-00405-g004].

Similar to the results presented in [Fig polymers-14-00405-g002], the standard deviation and error between triplicated results are very small. From the curve of best plot in [Fig polymers-14-00405-g004], it can be clearly observed that the adsorption has reached its maximum after 18 h of stirring. Below 18 h of stirring, the short contact time does not provide sufficient contact opportunity between the HPAM molecules and the CaCO_3_ particles. With increasing stirring time, the HPAM molecules have longer to adsorb onto the CaCO_3_ particles until a plateau is reached in the amount of adsorbed HPAM as a function of polymer concentration; this plateau is weakly dependent upon further increases in the stirring time. Hence, from [Fig polymers-14-00405-g004], a minimum HPAM–CaCO_3_ contact time of 18 h is required for maximum amount of HPAM adsorption onto CaCO_3_.

The HPAM–CaCO_3_ mixture was also subjected to stirring from 1 min to 6 h, and the supernatant was analysed with the Shimadzu UV2600 spectrophotometer. The data points from spectrometry were then fitted into different kinetic models to better understand the adsorption kinetics of HPAM onto CaCO_3_. The best-fitting kinetic model, which is the pseudo-second order model, is plotted in [Fig polymers-14-00405-g005], whereas the other two less-fitting models are shown in the Appendix A. The intraparticle diffusion model is also plotted in [Fig polymers-14-00405-g006].

The pseudo-first and pseudo-second order kinetic plot shows very high fitting, with pseudo-second order having the highest relation coefficient R^2^ of almost 1. This shows that the pseudo-second order model best describes the adsorption kinetics of the process, indicating that the adsorption of polyacrylamide onto carbonate particle surface is largely dependent on the adsorbate concentration. This is in line with the findings from Zhu et al. [16], whereby they found that the adsorption of polyacrylamide onto a quartz sand also displayed pseudo-second order kinetics, which suggested that the adsorption process is dependent on polymer concentration, and interaction between polyacrylamide and quartz sand is mainly through electrostatic and hydrogen bonding forces. Similarly, in the case of calcium carbonate, as the CO^−^ group of HPAM is responsible for electrostatic attraction with the surface Ca^2+^ ions of CaCO_3_ surface [16,30,31]. As for the intraparticle diffusion model in [Fig polymers-14-00405-g006], the plot shows a good degree of fitting, but it does not pass through the origin of the plot, indicating that intraparticle diffusion is not the rate-limiting step in the adsorption process. This finding is in line with that reported by Lv et al. [22], whereby the intraparticle diffusion of polyacrylamide is a rapid process, thus it cannot be the rate-limiting step.

### 3.3. Isothermal Adsorption Model Analysis

To better understand the adsorption isotherm, the adsorption data were fitted into a Langmuir, Freundlich and Temkin isotherm. The relation coefficients of each adsorption model are summarized in Appendix A, and the relevant adsorption information of the Langmuir isotherm is shown in Appendix A. The list of *R_L_* over initial concentration for each type of HPAM is tabulated in Appendix A. All this information is found in the Appendix A. The Langmuir adsorption model fitting results for F3330S, F3530S and F3630S are shown in [Fig polymers-14-00405-g007], while the other Freundlich and Temkin models are shown in the Appendix A.

From [Fig polymers-14-00405-g007], the Langmuir adsorption isotherm shows an extremely good fit to the three HPAM solutions, with relation coefficient R^2^ at approximately 0.97 and above (refer to Appendix A). The same is not observed in the Freundlich and Temkin adsorption isotherm in Appendix A where they only present a relation coefficient R^2^ of between 0.65–0.89 and 0.85–0.93, respectively. This finding agrees with results from different literature, stating that polyacrylamide adsorbs uniformly onto the surface of the adsorbent in the form of a monolayer [1,17,18,19], which is often best described by the Langmuir isotherm. The HPAM molecules are adsorbed homogenously on distinct localized sites of the CaCO_3_ with complete coverage of these sites [32]. This could explain why the Freundlich and Temkin models do not describe HPAM adsorption well, as these two isotherms are often used to describe adsorption on a heterogenous surface.

From Appendix A, it is observed that the maximum HPAM adsorption capacity per unit surface area CaCO_3_ (*Q_m_*) is found at F3630S, which is the HPAM with highest molecular weight among the three. However, the Langmuir constant (*K_L_*), which reflects the affinity between adsorbent and adsorbate, is found to be the highest for F3530S [33]. These data show good agreement with the adsorption results obtained from [Fig polymers-14-00405-g002], as the plot does reflect highest adsorption when F3630S is used, and the *Q_m_* is close to the plateau values in [Fig polymers-14-00405-g002].

From the tabulated *R_L_* values in Appendix A, all *R_L_* values for each polymer is between 0 and 1 regardless of polymer concentration, and the *R_L_* values decrease towards 0 as initial polymer concentration increases. This indicates that the adsorption of polyacrylamide onto calcium carbonate is a favorable adsorption process becoming increasingly favourable with rising polymer concentration. Amongst the three HPAM, F3530S shows the lowest overall *R_L_*, which is in line with the highest *K_L_* by F3530S when compared with the lower molecular weight polymers, which shows that this particular molecular weight HPAM has the highest affinity and adsorption favor with CaCO_3._

### 3.4. Zeta Potential Analysis

Zeta potential analysis is useful to understand the stability of the colloidal system. Before analysing the zeta potential of the mixture, the zeta potential of CaCO_3_ used was measured and the value was found to be +12.32 mV. Each measurement was triplicated, and the zeta potential plot of mixture using HPAM of different molecular weight is shown in [Fig polymers-14-00405-g008].

Based on the zeta potential results from [Fig polymers-14-00405-g008], all three HPAM–CaCO_3_ mixtures showed a decrease in zeta potential with increasing polymer concentration until a plateau in zeta potential at approximately 100 ppm for F3330S, while approximately 50 ppm for F3530S and F3630S. In the absence of HPAM, steric forces between CaCO_3_ particles induce repulsive forces between them. As HPAM was introduced, the zeta potential decreases with increasing HPAM concentration. A transition from positive to negative charge is observed approximately between 10 to 30 ppm, in which the pH of the solution is between 9–10. This is in line with the point of zero charge (PZC) of 9.40–9.60 as reported [34,35,36]. When no HPAM is present, the repulsive forces between the carbonate particles are dominated by electrical double layer repulsion, although the zeta potential is low so that van der Waals attractive forces are likely to dominate. As more HPAM molecules are introduced, the anionic HPAM molecules will adsorb onto the positively charged carbonate particles, thus gradually neutralising the carbonate surface charge and weakening the repulsive force. After the charge reversal point, continuing to increase the HPAM dosage only builds up negative charge, which causes the force to become repulsive again. However, this repulsive force is rather weak and can be easily overcome by interparticle bridging due to polymer adsorbing simultaneously on two particles. At approximately 100 ppm for F3330S and 50 ppm for F3530S and F3630S, the surface is said to be saturated with HPAM and the repulsive forces no longer increase [37]. Consequently, the plateau zeta potential becomes more negative with increasing polymer molecular weight. This agrees with the adsorption experiment which shows that adsorption of HPAM increases with higher polymer molecular weight. The higher HPAM concentration on CaCO_3_ surface induces slightly stronger anionic charge due to the carboxyl group of the polymer structure, which is then reflected as slightly more negative charge in zeta potential.

## 4. Conclusions

Hydrolysed polyacrylamides (HPAM) with different molecular weight were adsorbed onto calcium carbonate (CaCO_3_) samples, and data points from spectrometry are fitted into different adsorption isotherm and kinetic models. The overall HPAM adsorption onto CaCO_3_ seems best described by a Langmuir isotherm which suggests a monolayer adsorption onto a homogeneous surface. Tabulated *R_L_* values show favorable adsorption of HPAM onto CaCO_3_, with increasing values as the initial HPAM concentration increases. The adsorption also follows a pseudo-second order kinetics indicating that the HPAM interaction with CaCO_3_ is largely dependent on the adsorbate concentration. The adsorbed amount of F3330S and F3530S onto CaCO_3_ increases with increasing HPAM concentration until it reaches a plateau at 150 ppm, which indicates the saturation of the free-binding sites on CaCO_3_. The same is seen for F3630S from 300 ppm onwards. The study of the different molecular weights indicates that highest equilibrium adsorbed amount of HPAM is achieved by F3630S, which is the HPAM with highest molecular weight among the three. This is due to the fact that low molecular weight polymers tend to lie flat on a particle surface, thus occupying a larger fraction of surface which hinders late-coming polymers from attaching onto the surface. Lastly, the zeta potential analysis suggests that the overall charge of the colloidal system transitions from positive to a negative charge when HPAM is introduced into a CaCO_3_ colloidal system.

## Figures and Tables

**Figure 1 polymers-14-00405-g001:**
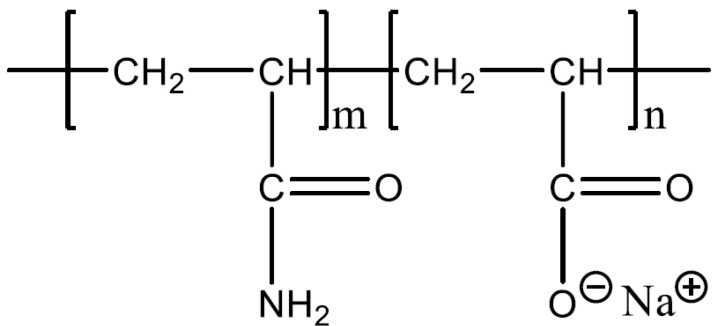
Molecular formula of HPAM.

**Figure 2 polymers-14-00405-g002:**
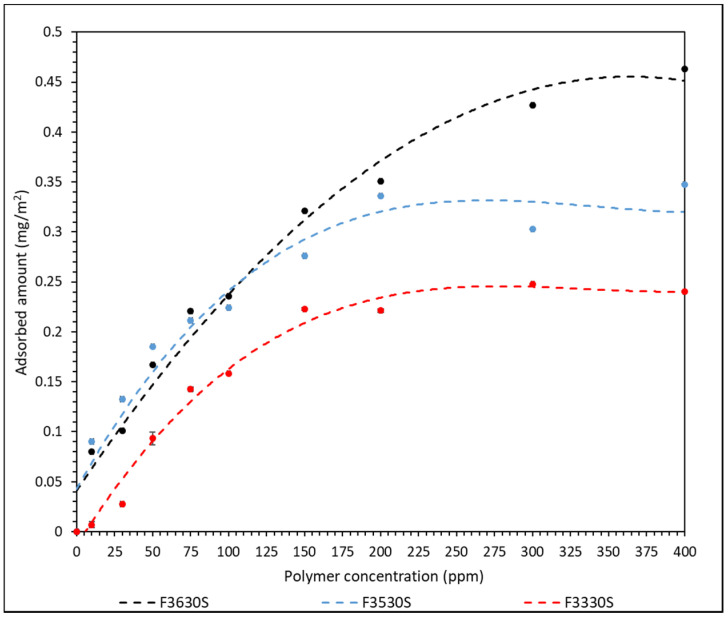
Amount of F3330S (11–13 MDa), F3530S (15–17 MDa) and F3630S (18–20 MDa) adsorbed onto CaCO_3_ against concentration of polymer dosage (Lines are a guide to the eye).

**Figure 3 polymers-14-00405-g003:**
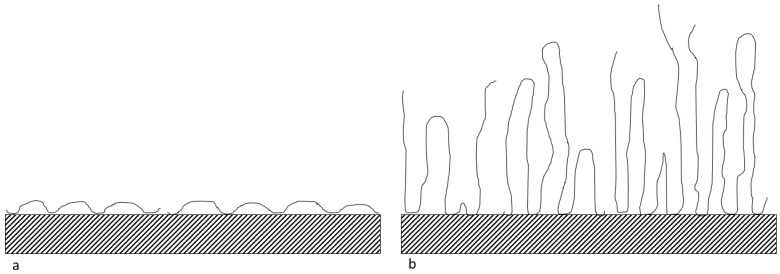
Configuration of (**a**) low molecular weight HPAM and (**b**) high molecular weight HPAM on the CaCO_3_ surface.

**Figure 4 polymers-14-00405-g004:**
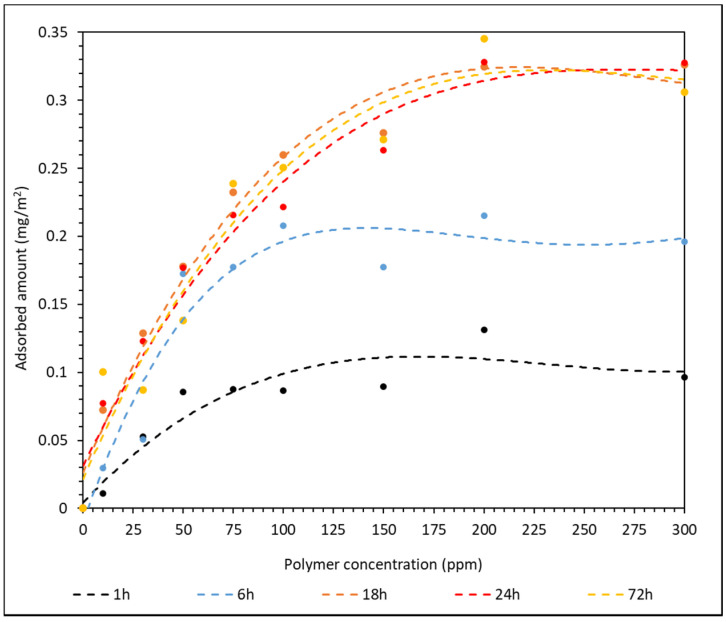
Effect of the stirring time on the amount of F3530S adsorbed onto CaCO_3_.

**Figure 5 polymers-14-00405-g005:**
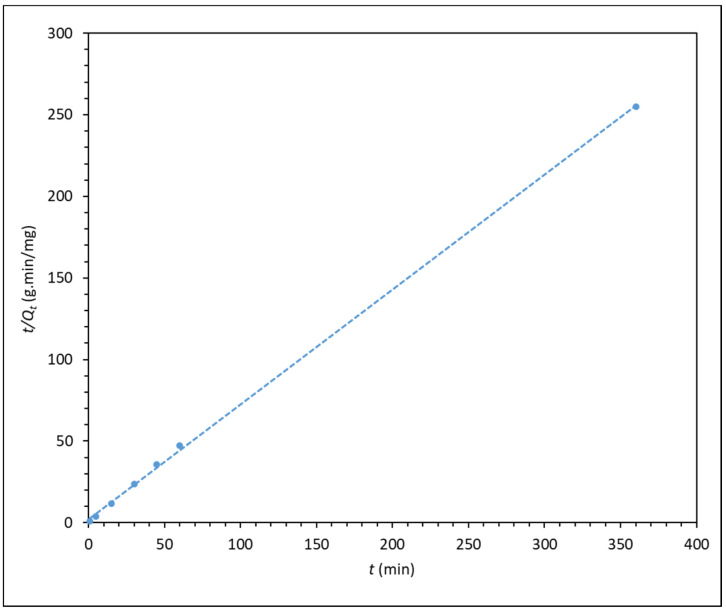
*t*/*Q**_t_* versus time (pseudo-second order kinetic model) for the adsorption of 300 ppm F3530S onto CaCO_3_. (R^2^ = 0.9996).

**Figure 6 polymers-14-00405-g006:**
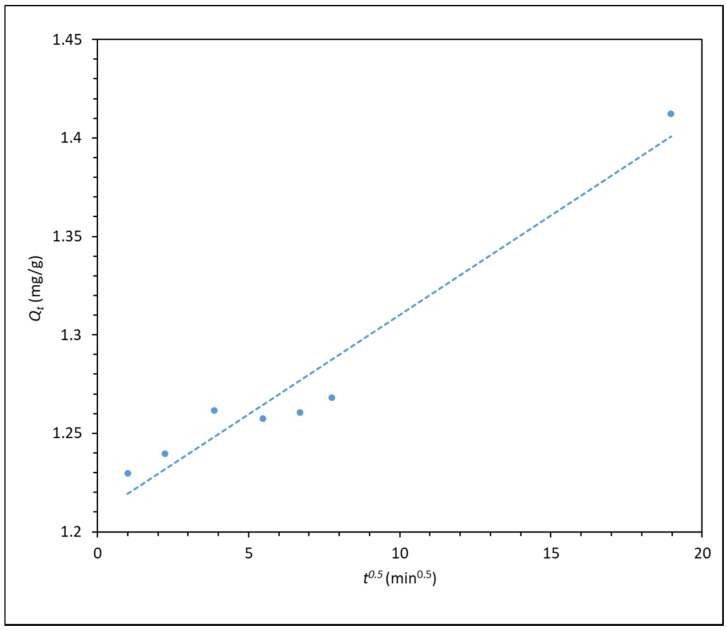
*Q_t_* versus *t*_0.5_ (intraparticle diffusion kinetic model) for the adsorption of 300 ppm F3530S onto CaCO_3_. (R^2^ = 0.949).

**Figure 7 polymers-14-00405-g007:**
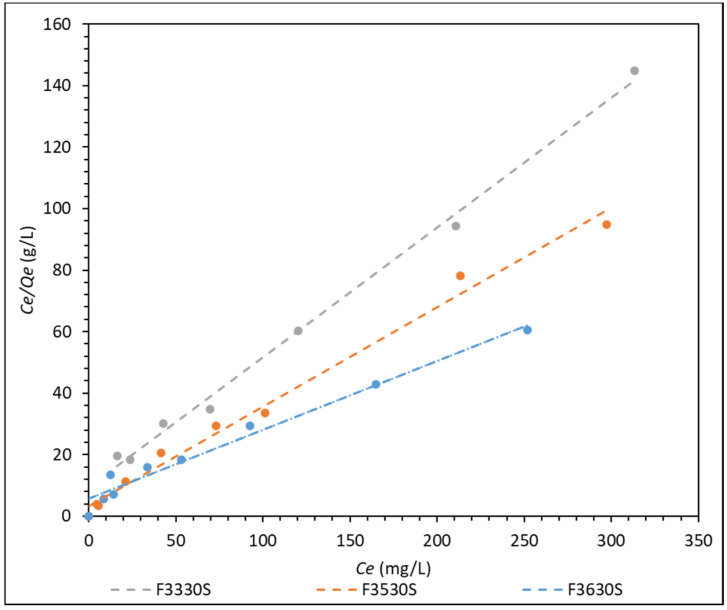
Langmuir adsorption isotherm model of F3330S, F3530S and F3630S adsorption onto CaCO_3_.

**Figure 8 polymers-14-00405-g008:**
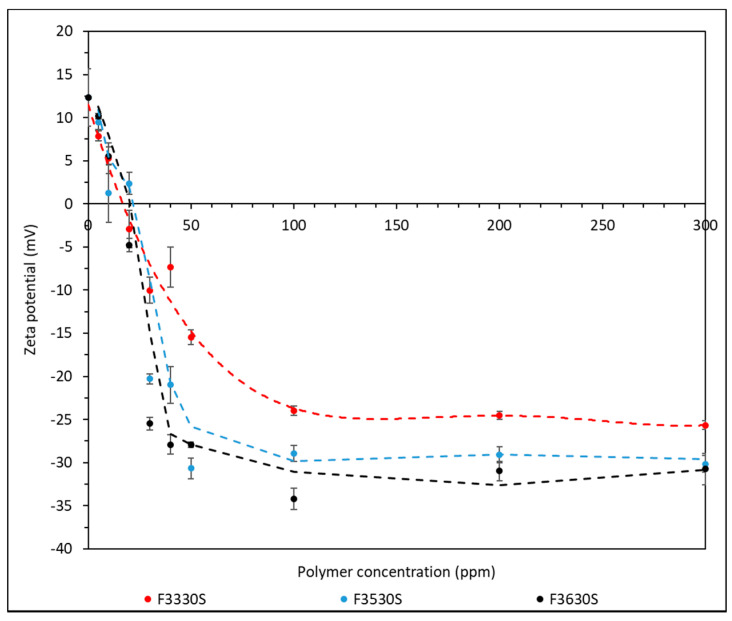
Zeta potential result of CaCO_3_ adsorbed with F3330S, F3530S and F3630S.

## Data Availability

Not applicable.

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
