# Peer review of "Adsorption of Hydrolysed Polyacrylamide onto Calcium Carbonate"

_polymers, 2022, doi:10.3390/polym14030405_

Round 1

Reviewer 1 Report

In their research article titled “Adsorption of hydrolyzed polyacrylamide onto calcium car- 2 Bonate” Jin Hau Lew and co-workers have investigated the three hydrolyzed polyacrylamides (HPAM) of different molecular weight (F3330S, 11-13MDa; F3530S, 15-17MDa; F3630S, 18-20MDa;) were adsorbed onto calcium carbonate (CaCO3) particles, and the results compared to different adsorption isotherm and kinetic models.  In addition, HPAM adsorption onto CaCO3 is best described by Langmuir  isotherm  with  RL  value of between  0  and  1,  which  suggests favorable  monolayer  adsorption of  HPAM onto CaCO3. The authors perform good quality work for the scientific community, and the authors explain the results with a mechanism clearly. I appreciate their efforts; however, modify your manuscript according to the following changes. I think it could be published with minor revisions.

Q 1, please also add some quantitative data in the abstract.

Q 3, to understand the effect of HPAM concentration on its adsorption onto CaCO3, 1g of 

CaCO3 was added into three types of HPAM, namely F3330S (11 – 13 MDa), F3530S (15 – 17 MDa), and F3630S (18–20 MDa) at different concentrations, and the adsorption was investigated by determining the concentration of polymer remaining in the solution by measuring the absorbance of the supernatant using the Shimadzu UV2600 spectrophotometer. Move this part to the experiment section, then the result and discussion. 

Q 4, The calibration curves for each polymer are shown in line 182 reference is missing and adds the supplementary material, for example, figure number table number.

Q 5, In Figures 2 and others, the inferior graphic quality needs to improve and follow the same stander for all representation throughout the manuscript.

Q 6, Lower molar mass polymer tends to adopt a flat configuration, with each section of molecules occupying a more significant fraction of the adsorbent surface. In the case of high molecular weight polymer, there is a higher likelihood of loops and tails formation on the particle surface; hence the polymers are not lying flat on the surface. The increased coverage of low molecular weight Polymers 2021, 13, x FOR PEER REVIEW  6  of  18 polymers is more effective in hindering late-coming polymers than high molecular  220 weight polymers. Would you please explain this phenomenon by adding suitable fig or scheme?

  • Q 7, as the author showed in figure 3, the adsorption has reached its maximum after 18h of stirring. But he does not explain the initial absorption phenomena. Would you please explain the initial phenomena of the adsorption? Such as explain in the following articles. 3390/polym13020268  and 10.1039/D1RA90157E.

Q 8, Are these references in proper format? For example,

  • Aggour, M.A.; Abu Khamsin, S.A., and Osman, E.S.A. A New method of Sand Control: The Process and Its First Field Implementation. in SPE/IADC Middle  East  Drilling and  Technology    Amsterdam,The  Netherlands  2007. Society of Petroleum Engineers.
  • Lee, R.Y. Development of sand agglomeration formulation for oil & gas well applications to reduce the production of fine particulates. Doctoral dissertation, Imperial College London, London, June 2019.
  • Nouri, A.; Vaziri, H.; Belhaj, H., and Islam, R. Effect of Volumetric Failure on Sand Production in Oil-Wellbores. SPE 80448. in SPE - Asia Pacific Oil and Gas Conference and Exhibition. Jakarta, Indonesia 2003.

Author Response

We would like to thank the reviewer for the swift reviewing process and for the questions/comments made. We have adjusted the manuscript to take these into account and now feel that the manuscripts is much improved

Q 1, please also add some quantitative data in the abstract.

Response to Q1: quantitative data are added into abstract

Please note there was no question 2

Q 3, to understand the effect of HPAM concentration on its adsorption onto CaCO3, 1g of

CaCO3 was added into three types of HPAM, namely F3330S (11 – 13 MDa), F3530S (15 – 17 MDa), and F3630S (18–20 MDa) at different concentrations, and the adsorption was investigated by determining the concentration of polymer remaining in the solution by measuring the absorbance of the supernatant using the Shimadzu UV2600 spectrophotometer. Move this part to the experiment section, then the result and discussion.

Response to Q3: the cited paragraph is moved and incorporated to its corresponding section in section 2.2.2 line 150 - 151.

Q 4, The calibration curves for each polymer are shown in line 182 reference is missing and adds the supplementary material, for example, figure number table number.

Response to Q4: the missing references are added.

Q 5, In Figures 2 and others, the inferior graphic quality needs to improve and follow the same stander for all representation throughout the manuscript.

Response to Q5: graphic quality has been improved and all graphs are standardised.

Q 6, Lower molar mass polymer tends to adopt a flat configuration, with each section of molecules occupying a more significant fraction of the adsorbent surface. In the case of high molecular weight polymer, there is a higher likelihood of loops and tails formation on the particle surface; hence the polymers are not lying flat on the surface. The increased coverage of low molecular weight Polymers 2021, 13, x FOR PEER REVIEW 6 of 18 polymers is more effective in hindering late-coming polymers than high molecular 220 weight polymers. Would you please explain this phenomenon by adding suitable fig or scheme?

Response to Q6: an illustration of the phenomenon is added as Figure 3

Q 7, as the author showed in figure 3, the adsorption has reached its maximum after 18h of stirring. But he does not explain the initial absorption phenomena. Would you please explain the initial phenomena of the adsorption? Such as explain in the following articles. 10.3390/polym13020268 and 10.1039/D1RA90157E.

Response to Q7: an explanation is added as shown from line 291 to 296.

Reviewer 2 Report

Can be accepted after major revision

Author Response

Q1 Some information from 3.1. Polymer Adsorption Analysis should be re-moved page 4 Lines 177-182 to experimental part.

Response to Q1: the cited paragraph is moved and incorporated to its corresponding section in section 2.2.2 line 150 – 151.

Q2 Some information from 3.2. Polymer Adsorption Kinetics should be re-moved page 6 Lines 240-243 to experimental part.

Response to Q2: the cited paragraph is amended, and the paragraph is incorporated to its corresponding section in section 2.2.2 line 157 – 158

Q3 The equations of kinetic and isothermal models from the results and discussion sections should be re-moved to the experimental sector. Or there should be a separate sector − theoretical section.

Response to Q3: the paragraphs on kinetic and isothermal models were transferred to section 2.2.3 and section 2.2.4.

Please note that we have also made some very minor improvements to the readability and formatting of the manuscript which were not suggested by the two referees

Round 2

Reviewer 2 Report

Can be accepted in present form